# Cardiovascular disease subtypes, physical disability and workforce participation: A cross-sectional study of 163,562 middle-aged Australians

**Muhammad Shahdaat Bin Sayeed**○*, **Grace Joshy, Ellie Paige, Emily Banks, Rosemary Korda**

National Centre for Epidemiology and Population Health, Research School of Population Health, Australian National University, Canberra, ACT, Australia

* muhammad-shahdaat.bin-sayeed@anu.edu.au

## Abstract

### Background

Workforce participation is reduced among people with cardiovascular disease (CVD). However, detailed quantitative evidence on this is limited. We examined the relationship of CVD to workforce participation in older working-age people, by CVD subtype, within population subgroups and considering the role of physical disability.

### Methods

Questionnaire data (2006–2009) for participants aged 45–64 years (n = 163,562) from the population-based 45 and Up Study (n = 267,153) were linked to hospitalisation data through the Centre for Health Record Linkage. Prior CVD was from self-report or hospitalisation. Modified Poisson regression estimated adjusted prevalence ratios (PRs) for non-participation in the workforce in people with versus without CVD, adjusting for sociodemographic factors.

### Results

There were 19,161 participants with CVD and 144,401 without. Compared to people without CVD, workforce non-participation was greater for those with CVD (40.0% vs 23.5%, PR = 1.36, 95%CI = 1.33–1.39). The outcome varied by CVD subtype: myocardial infarction (PR = 1.46, 95%CI = 1.36–1.55); cerebrovascular disease (PR = 1.92, 95%CI = 1.80–2.06); heart failure (PR = 1.83, 95%CI = 1.68–1.98) and peripheral vascular disease (PR = 1.76, 95%CI = 1.65–1.88). Workforce non-participation in those with CVD versus those without was at least 21% higher in all population subgroups examined, with PRs ranging from 1.75 (95%CI = 1.65–1.85) in people aged 50–55 years to 1.21 (95%CI = 1.19–1.24) among those aged 60–64. Compared to people with neither CVD nor physical functioning limitations, those with physical functional limitations were around three times as likely to be out of the workforce regardless of CVD diagnosis; participants with CVD but without physical

**Data Availability Statement:** This study uses data from the 45 and Up Study, which is managed by the Sax Institute in collaboration with major partner

Cancer Council NSW, and partners: the National Heart Foundation of Australia (NSW Division); NSW Ministry of Health; NSW Government Family & Community Services–Ageing, Carers and the Disability Council NSW; and the Australian Red Cross Blood Service. Data supporting the findings from this study are available from the Sax Institute, the NSW Department of Health and the Australian Bureau of Statistics, with data linkage conducted by the NSW Centre for Health Record Linkage. Restrictions apply to the availability of these data, which were used under license for the current study, and so are not publicly available. Researchers may apply for access to these data with the appropriate data custodian and ethics approvals. Information about data access and governance policies is available at: https://www.saxinstitute.org.au/our-work/45-up-study/for-researchers/.

**Funding:** Muhammad Shahdaat Bin Sayeed is supported by an Australian Government Research and Training scholarship to undertake a PhD at the Australian National University, Canberra, ACT, Australia. Emily Banks is supported by a Principal Research Fellowship from the National Health and Medical Research Council of Australia (reference 1136128). Ellie Paige is supported by a Postdoctoral Fellowship (reference: 102131) from the National Heart Foundation of Australia. This project was supported by a Project Grant from the National Health and Medical Research Council of Australia (reference 1139539).

**Competing interests:** The authors declare no other competing interests.

functional limitations were 13% more likely to be out of the workforce (PR = 1.13, 95%CI = 1.07–1.20).

## Conclusions

While many people with CVD participate in the workforce, participation is substantially lower, especially for people with cerebrovascular disease, than for people without CVD, highlighting priority areas for research and support, particularly for people experiencing physical functioning limitations.

## Introduction

Cardiovascular disease (CVD) remains a leading contributor to global burden of disease internationally [1] and is the second largest contributor to burden of disease in Australia [2], despite declining CVD mortality globally. With improving CVD survival, there is an increasing need to consider the consequences of living with CVD, for individuals and society. Many people with CVD are disabled to the extent that their core activities, including self-care, mobility, and communication are affected [3, 4], as is their ability to engage socially and participate in the workforce [5]. The effect of CVD on workforce participation is of particular importance. It not only affects the overall health and financial well-being of the person living with CVD [6], but also has consequences for society given the substantial economic benefits of retaining people in the workforce [7], an increasingly important issue as the population ages.

Studies outside Australia, primarily set in Europe, have shown that CVD is associated with poorer workforce outcomes, including higher rates of unemployment, exit from paid employment, receipt of disability pension and early retirement [8–13]. In Australia, studies have reported lower productivity in work [14], lower income [15–17] and higher retirement due to ill-health [18] among people with CVD compared to those general population, but there is no large-scale comparative evidence on workforce participation in people with and without CVD. Furthermore, CVD is heterogeneous and there is a lack of comparative evidence on workforce participation across with different types of CVD and in different population subgroups. There is also limited information on why people with CVD may be less likely to participate in the workforce. In particular, although physical functional limitations are common among people with CVD [19] and are a key determinant of employment [20], no studies have examined the extent to which this might account for the lower workforce participation among people with CVD.

In this study we aimed to quantify workforce participation in a large Australian population-based study of 45-64-year-old men and women, comparing levels of non-participation in the workforce in people with versus without CVD, overall and according to CVD subtypes including ischaemic heart disease (IHD) and its subgroup myocardial infarction (MI), cerebrovascular disease, heart failure (HF) and peripheral arterial disease (PAD). We also aimed to examine whether the relationship between CVD and workforce participation varied by socio-demographic and health factors, and in particular, the extent to which co-existing physical functioning limitations–as a measure of physical disability–might explain differences in workforce participation between people with and without CVD. Such evidence is likely to be useful to inform CVD survivors, their care, policy and practice and in modelling CVD outcomes and costs.

## Materials and methods

### Study population and data sources

We used data from the Sax Institute's 45 and Up Study, a population-based study of 267 153 people aged 45 and over in New South Wales (NSW) (~10% of the population in that age group), randomly sampled from the Medicare Australia database. Individuals joined the study by completing a postal questionnaire (available at https://www.saxinstitute.org.au/our-work/45-up-study/questionnaires/) and providing consent for follow-up through linkage to a range of routinely-collected data. The majority of participants were sampled in 2008 and the median baseline questionnaire date is February 2008. Persons aged 80 years and over and those living in rural and remote areas were oversampled by a factor of two. The response rate to mailed invitations was estimated to be 18%, representing around 10% of the NSW population aged 45 years and older [21]. This study used cross-sectional data from the baseline survey which included self-reported data from participants on socio-demographic, family, work and health. We also used linked data from the NSW Admitted Patient Data Collection to identify CVD diagnoses prior to study enrolment. The Admitted Patient Data Collection is a complete census of patients admitted to hospitals in NSW (available from 2001), covering public and private hospitals and private day procedure centres, and includes information on diagnosis, and procedures. The 45 and Up Study data and administrative health datasets were probabilistically linked through the Centre for Health Record Linkage (CHeReL) with false positive and negative rates of <0.5% and <0.1%, respectively. The participants with invalid data on age or date of recruitment (n = 454), those with linkage errors (n = 195) and those aged 65 years or over (n = 102,942) were excluded. There were 163,562 participants who were of working age (45-<65 years old) with known CVD status at baseline (70 458 men, 93 104 female) (details in S1-S4 Figs, S1-S3 Tables in S1 File).

### Outcome: Workforce participation

The main outcome of interest was non-participation in paid work (yes/no). This was based on responses to the following two questions: "What is your current work status?" and "About how many hours each week do you usually spend doing the following—paid work, voluntary/unpaid work"?. Those indicating valid paid hours (> 0 and <100) or work status as at least one of "In full time paid work", "In part time paid work", "Self-employed", "Partially retired" were classified as participating in the workforce; of the remaining participants, those indicating work status as "Doing unpaid work", "Completely retired/pensioner", "Studying", "Looking after home/family", "Disabled/sick", "Unemployed", "Other" were classified as not participating in the workforce.

Secondary outcomes included paid work hours/week among those in paid work, retirement (yes/no), and retirement due to ill-health (yes/no) among those not in paid work. Paid work hours/week was defined based on responses to the following questions: "What is your current work status?" and "About how many hours each week do you usually spend doing the following?—paid work, voluntary/unpaid work". Zero or non-zero positive values less than 100 were considered valid paid hours per week. Further logical checks was applied with workforce participation status variable (S4 Table in S1 File).

Retirement was defined from question that asked 'If you are partially or completely retired, why did you retire?'. Participants with valid records for any of the options ("Reached usual retirement age", "Lifestyle reasons", "To care for family members/friend", "Ill health", "Made redundant", "Made redundant", "Could not find a job", "Other") were defined as 'retired' and those without any of these options were defined as 'not retired'. Following logical checks,

participants not in the work force were classified as "yes (retired due to ill health)" if they chose "ill health", and otherwise as "no (retired for other reasons)".

Further outcome specific exclusions for missing or invalid data applied for each outcome. Among the 163,562 participants included in the study, 131 had missing data on workforce participation status. Of the 121,816 participants in the work force, 7752 had missing/invalid data on the number of paid hours per week and of the 41,615 participants not in the work force, 11,961 had invalid data on retirement due to ill health (S1 Fig in S1 File).

## Main exposure: Cardiovascular disease

Participants were classified as either having CVD or not. CVD was defined as self-reported heart disease, stroke or blood clot on the baseline questionnaire, or at least one hospital admission in the five years prior to entering the study with a major CVD diagnosis code, as identified in any diagnostic or a procedure code fields in the linked hospital admissions data (details in S5 Table in S1 File) [22]. A five-year window was used to ensure uniform probability of identification of previous diagnoses from administrative data for all participants. We also sub-categorised participants based on hospitalisations for the following CVD subtypes (yes/no): IHD (ICD-AM codes: I20-I25), MI (ICD-AM codes: I21, I22 and I23), cerebrovascular disease (ICD-AM codes: I61, I63, I64), PAD (ICD-AM codes: I70-I74) and HF (ICD-AM codes: I50, I11.0, I13.0, I13.2). The 'other CVD' group consisted of those participants with CVD who had self-reported CVD or any CVD codes other than the CVD subtype as mentioned earlier. The participants with hospitalisation for a particular CVD subtype may or may not have had another types of CVD hospitalisations. In the sensitivity analyses, we further stratified people with CVD into self-reported CVD only, CVD hospitalisation only or both, and in combination of several CVD subtype where sample size permitted.

## Other variables of interest

Sociodemographic and health variables known to be associated with workforce participation [23–32], were used to define population subgroups. The sociodemographic variables included: age, sex, remoteness of residence (categorised as major cities, inner regional and more remote), marital status (categorised as married/defacto and single/widowed/divorced), education attainment (categorised as tertiary, certificates/diploma/trade and high school or less), language spoken at home other than English (LOTE) (yes/no) and born in Australia/New Zealand (yes/no). The health-related variables included: body mass index (BMI, kg/m$^2$) categorised as underweight (15-<18.5), normal weight (18.5-<25), overweight (25-<30), obese (30–50); alcohol consumption (number of alcoholic drinks per week categorised as non-drinkers (zero drinks per week), moderate drinkers (>0-<15 drinks per week), heavy drinkers (≥15 drinks per week)); smoking status (non-smoker, past-smoker, current smoker); self-reported doctor-diagnosed diabetes/cancer/osteoarthritis (yes/no for each) and physical functioning limitations. The cut-points of alcohol consumption broadly reflect Australian guidelines on low-risk consumption [33]. Physical functioning limitations were assessed using the Medical Outcomes Study–Physical Functioning (MOS-PF) subscale which was based on 10 questionnaire items assessing varying levels of physical functioning [34]. Physical functioning limitations scores ranged from 0 to 100, with higher scores representing fewer limitations, and were grouped into four categories: no limitation (score of 100); minor limitation (score 90–99); moderate limitation (60–89); and severe limitation (score 0–59) [35] (details in S6 Table in S1 File). These variables were based on the self-reported data from the baseline survey (2006–2009) of the 45 and Up Study, except for sex (obtained from Medicare Australia database [21])

and remoteness of residence (derived from the mean Accessibility Remoteness Index of Australia Plus score [36]).

## Statistical analysis

Descriptive statistics were used to summarise characteristics of the study population and distribution of outcomes by CVD status. Modified Poisson regression with robust error variance [37] was used to estimate prevalence ratios (PRs) for non-participation in the paid workforce in relation to CVD. Models were sequentially adjusted, initially adjusting for age-group (5-year age bands) and sex [model 1], noting no sex by CVD interaction was observed, and then additionally adjusted for region of residence and education [model 2]. Further statistical adjustments were not done as the objective was to compare prevalences and lived experiences; causality cannot be established in this cross-sectional analysis. We also estimated PRs separately within population subgroups; chi-square tests for heterogeneity were used to assess heterogeneity between subgroups. To examine the potential contribution of physical functioning to the CVD-workforce participation relationship, we modelled the joint categorisation of CVD and physical functioning limitations on workforce participation; those with no CVD and no physical functioning limitations were used as the reference group. Participants with missing values for workforce participation were excluded from all analyses. There were no missing data in the main exposure (CVD status), age or sex. Missing values for the factors used in the model adjustments were included in the analysis as separate categories.

Self-reported CVD and hospital recorded CVD was considered separately in the first sensitivity analysis, the hospital recorded CVD subtypes were considered after excluding participants with multiple CVD subtypes in the second sensitivity analysis, and combination of more than one CVD subtypes in the third sensitivity analysis.

In supplementary analysis, the PRs for retirement and retirement due to ill health were estimated using similar methods as those for non-participation in paid workforce, and a generalised linear model assuming a Poisson distribution and log link function was used to estimate mean of paid work hours per week.

Analyses was carried out using SAS software version 9.4 and R version 3.5.2 [38].

## Ethics approval

Individuals gave written informed consent to take part in the study, including consent for linkage of their data to population health databases. The conduct of the 45 and Up Study was approved by the University of New South Wales Human Research Ethics Committee (HREC). Ethics approval for this project was obtained from the NSW Population and Health Services Research Ethics Committee (Reference: HREC/12/CIPHS/31) and the Australian National University Human Research Ethics Committee (Reference: 2012/504).

## Results

### Characteristics of the study participants

There were 163,562 study participants: 19,161 (11.7%) with CVD and 144,401 (88.3%) without CVD. 121,816 participants (74.5%) were in the paid workforce, of whom 11,480 (9.4%) had CVD; and 155,723 participants (>95%) had valid paid work hours per week (i.e. $\geq 0$ and $<100$). There were 43,397 participants who had retired, 29,654 of retirees had not been working in any form and 5,970 of whom had CVD (20.1%). The sociodemographic profile of participants with and without CVD was similar except that the CVD group had higher proportions of men and older participants (**Table 1**). Participants with CVD had a poorer health profile

**Table 1. Sociodemographic and health related characteristics of study participants.**

| | People with CVD | People without CVD |
|---|---|---|
| Total participants (n = 163562) | 19161 | 144401 |
| Percentage (%) | 11.7 (19161/ 163562) | 88.3 (144401/ 163562) |
| Age (years) | | |
| Mean (sd) | 57.5 (5.18) | 55.0 (5.38) |
| 45–49 | 11.0 (2110) | 22.7 (32759) |
| 50–54 | 19.7 (3767) | 27.1 (39173) |
| 55–59 | 29.8 (5712) | 27.5 (39733) |
| 60–64 | 39.5 (7572) | 22.7 (32736) |
| **Sex** | | |
| Men | 51.5 (9873) | 42.0 (60585) |
| Women | 48.5 (9288) | 58.0 (83816) |
| **Region of residence** | | |
| Major cities | 49.5 (9477) | 51.9 (75002) |
| Inner regional | 36.1 (6911) | 34.7 (50058) |
| More remote | 12.4 (2370) | 11.4 (16392) |
| **Marital status** | | |
| single/widowed/separated | 23.1 (4425) | 20.1 (29042) |
| Married/defacto | 76.3 (14613) | 79.3 (114549) |
| **Education attainment** | | |
| No school certificate | 12.8 (2460) | 7.8 (11225) |
| Certificate/diploma/trade | 64.1 (12277) | 62.3 (89897) |
| Tertiary | 22.0 (4206) | 29.0 (41853) |
| **Language spoken at home other than English (Yes)** | 8.5 (1623) | 10.0 (14443) |
| **Country of birth in Australia/NZ** | 79.6 (15244) | 78.0 (112655) |
| **Alcohol consumption** | | |
| None (0 drink per week) | 34.5 (6604) | 28.9 (41680) |
| Moderate drinkers (1-14drinks per week) | 48.2 (9237) | 54.6 (78857) |
| Heavy drinkers (15 or more drinks per week) | 15.6 (2988) | 15.1 (21793) |
| **Smoking status** | | |
| Current | 10.7 (2046) | 9.3 (13367) |
| Past | 40.9 (7839) | 33.0 (47590) |
| Never | 48.0 (9194) | 57.5 (82979) |
| **BMI Category** | | |
| Underweight (15-<18.5) | 0.9 (169) | 0.9 (1369) |
| Normal weight (18.5-<25) | 24.3 (4659) | 34.7 (50047) |
| Overweight (25-<30) | 36.3 (6959) | 36.1 (52120) |
| Obese (30 to 50) | 31.3 (6005) | 21.6 (31159) |
| **Medical History: Cancer: Yes** | 15.2 (2919) | 10.7 (15416) |
| **Medical History: Diabetes: Yes** | 14.4 (2753) | 5.4 (7804) |
| **Medical History: Osteoarthritis: Yes** | 4.8 (921) | 2.8 (4026) |
| **Physical functioning limitations** | | |
| No limitation | 22.7 (4347) | 41.7 (60166) |
| Minor limitation | 24.6 (4714) | 26.7 (38603) |
| Moderate limitation | 24.2 (4645) | 16.1 (23275) |
| Severe limitation | 18.5 (3536) | 6.1 (8812) |

Note: % missing responses (CVD, No CVD): region of residence (2.1, 2); marital status (0.6, 0.6); education attainment (1.1, 1), country of birth (0.8, 0.8), alcohol per week (1.7, 1.4); smoking status (0.4, 0.3); BMI (7.1, 6.7); physical functioning limitations (10.0, 9.4). sd refers to standard deviation, NZ refers to New Zealand, BMI refers to Body Mass Index.

than those without CVD, with higher levels of smoking, obesity, comorbid diseases and moderate/severe functional limitation (**Table 1**).

## CVD and workforce participation

Overall, 60% of people with CVD were participating in the workforce, compared to 76% of people without CVD (**Table 2**). Mean weekly working hours per week among those in paid work was slightly lower overall for people with CVD than without CVD. Of all participants, 42% of people with CVD had retired, compared to 25% of people without CVD. Of the retirees who were not in paid workforce in any form, 53% of people with CVD had retired due to ill health, compared to 26% of people without CVD. In every 5-year age bracket (45-<50 to 60-<65), non-participation was higher, weekly work hours slightly lower, retirement and retirement due to ill health higher in men and women with CVD than without CVD (**Fig 1**).

**Table 2. Workforce participation, paid work hours per week, retirement patterns and physical functioning limitations among study participants.**

| | People with CVD | People without CVD |
|---|---|---|
| **Total N** | 19161 | 144401 |
| **In workforce**[*] | **59.9** (11480) | **76.4** (110336) |
| In full time paid work | 28.0 (5358) | 38.8 (55971) |
| Self-employed | 14.0 (2675) | 17.3 (24913) |
| In part time paid work | 15.4 (2950) | 19.7 (28512) |
| Partially retired | 6.8 (1297) | 5.3 (7644) |
| **Not in workforce** | **40.0** (7657) | **23.5** (33958) |
| Doing unpaid work | 5.8 (1102) | 5.3 (7681) |
| Completely retired/pensioner | 21.5 (4113) | 12.3 (17729) |
| Studying | 1.7 (319) | 2.1 (3053) |
| Looking after home/family | 9.9 (1902) | 11.2 (16182) |
| Disabled/sick | 14.9 (2859) | 4.4 (6410) |
| Unemployed | 3.8 (732) | 3.1 (4502) |
| Other | 2.0 (380) | 1.7 (2419) |
| **Paid hours of work** (for those in workforce) | | |
| N | 10506 | 103558 |
| Mean, SD | 34.9, 15.6 | 35.9, 14.8 |
| Median [inter quartile range] | 38 [25, 45] | 38 [26, 45] |
| **Retirement reasons** (among those who retired and not in the workforce) | | |
| **Total N** | 5970 | 23684 |
| **Retired due to ill health** | 53.0 (3166) | 26.3 (6238) |
| **Retired due to other reasons** | 47.0 (2804) | 73.7 (17446) |
| Reached usual retirement age | 7.9 (474) | 12.2 (2896) |
| Lifestyle reasons | 20.7 (1237) | 32.4 (7684) |
| To care for family member/friend | 10.9 (648) | 14.9 (3537) |
| Made redundant | 10.6 (630) | 11.7 (2770) |
| Could not find a job | 5.6 (332) | 5.4 (1277) |
| Other | 9.8 (584) | 14.1 (3328) |

One person might be in more than one category of sub-groups of those in workforce, not in workforce and retired due to other reasons.

[*]% missing responses of workforce participation (Total, CVD, No CVD): (0.08, 0.13, 0.07).

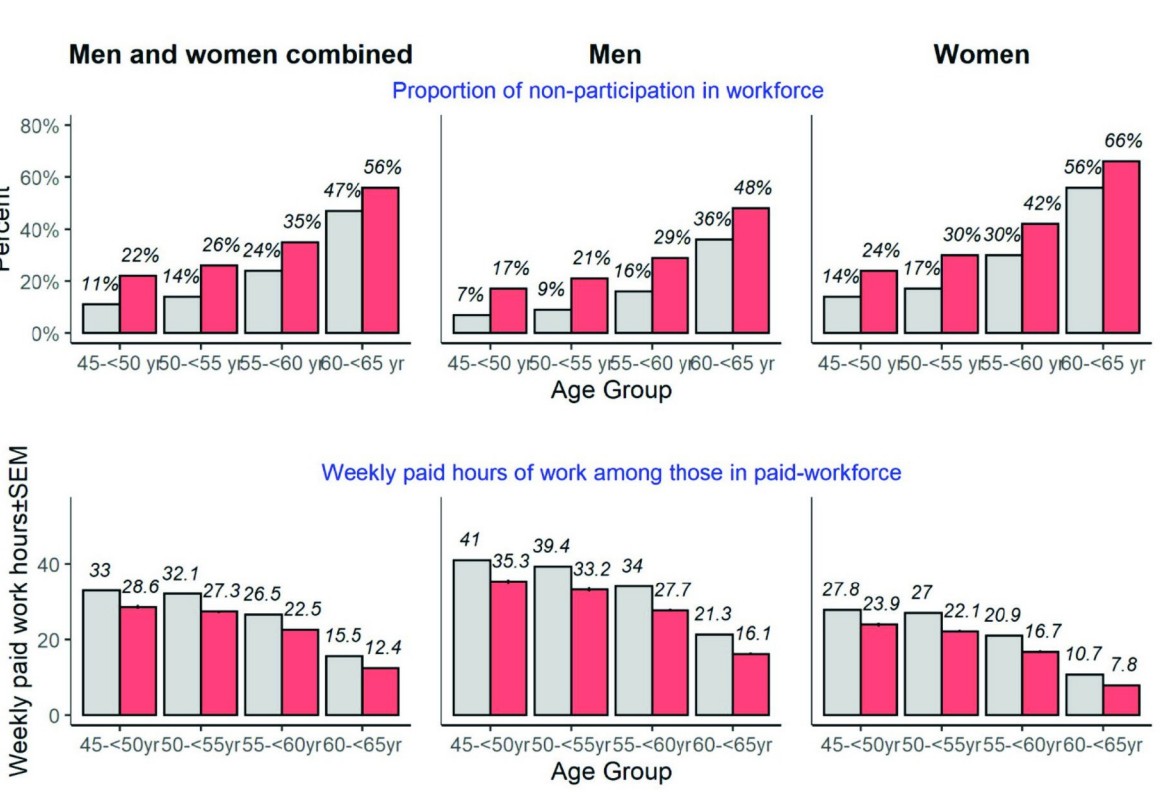

SEM=Standard error of mean, and SEM might not be always visible becasue of relatively smaller values

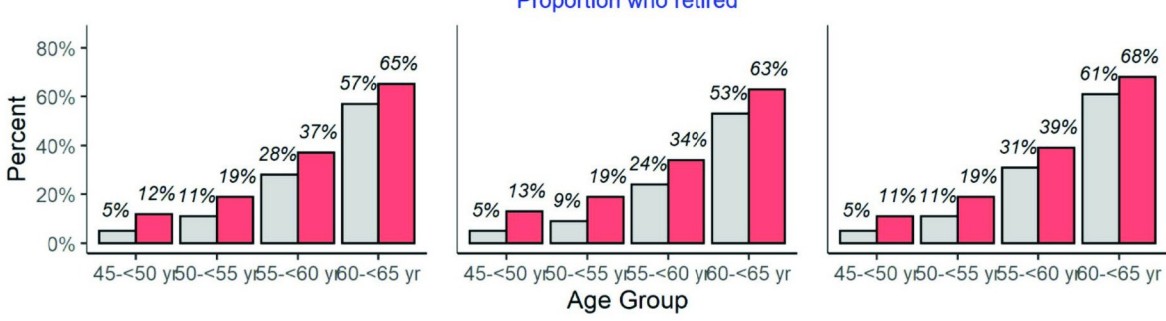

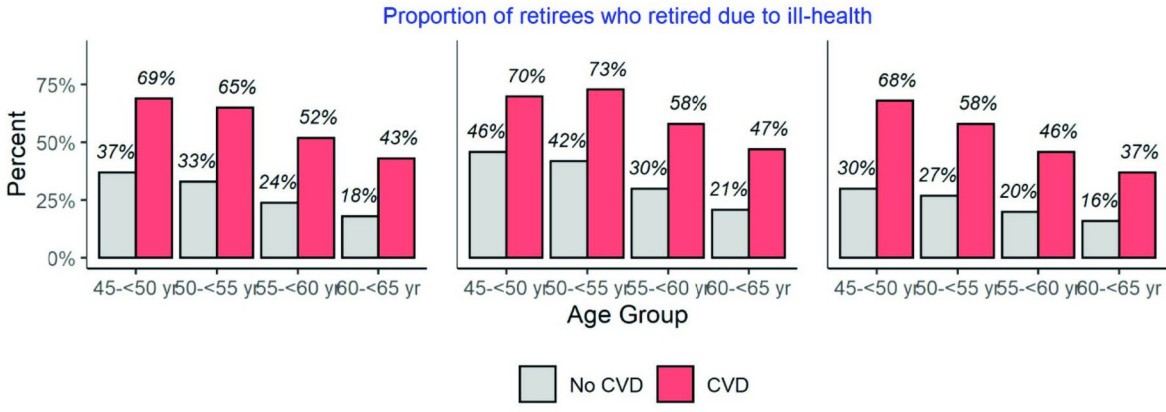

**Fig 1. Workforce participation characteristics among study participants according to CVD status, sex and age-group.**

After adjusting for sociodemographic characteristics (age, sex, region of residence and education), workforce non-participation was 36% higher among people with any CVD compared to people without CVD [prevalence ratio (PR) = 1.36 (95% CI: 1.33–1.39)]. Workforce non-participation varied by CVD subtype, with PRs of 1.46 (95% CI: 1.41–1.50) for IHD and 1.92 (95% CI: 1.80–2.06) for cerebrovascular disease (**Fig 2**). Sensitivity analysis indicate that those with CVD hospitalisation had somewhat higher PRs of non-participation in the paid workforce compared to those with self-reported CVD only (S7 Table in S1 File) and people with only one type of CVD had slightly lower PRs of non-participation in the paid workforce than those with more than one type of CVD (S8 and S9 Tables in S1 File). A similar pattern of results was seen for work hours among those who are in paid workforce, retirement among all participants and retirement due to ill health of retirees who had not been working in any form (S5-S7 Figs, S10-S18 Tables in S1 File).

## CVD and workforce participation among population subgroups

Workforce participation was associated with a number of sociodemographic factors among both people with and without CVD, including age, marital status, education, country of birth, alcohol consumption, smoking status and having medical history of osteoarthritis (Fig 3, S8 and S9 Figs in S1 File). When workforce participation was compared in people with and without CVD within sociodemographic and health subgroups, workforce non-participation remained higher among people with CVD compared to those without CVD, regardless of population subgroup. However, PRs were significantly higher among younger people, men, those who were not married/de facto, those without tertiary qualifications and those who were current smokers. Although the absolute crude prevalence of workforce non-participation was higher in older compared to younger age groups (irrespective of CVD status), the relation of CVD and workforce non-participation became weaker with increasing age (**Fig 3**).

## CVD, physical functioning limitations and workforce participation

Workforce participation was lower in those with greater physical functional limitations—among both those with and without CVD—but non-participation was higher among those with CVD in all sub-groups based on physical functioning limitations (**Fig 4**). Among participants with no physical functioning limitations, about one in 5 were not working—21% of those with CVD and

| | % (Not in workforce/Total) | Model[1] PR (95% CI) | Model[2] PR (95% CI) | |
|---|---|---|---|---|
| All participants | 25.5 (41615/163431) | | | |
| Any CVD [a] | 40.0 (7657/19137) | 1.43 (1.40-1.46) | 1.36 (1.33-1.39) | |
| *Ischaemic heart disease* [b] | 43.0 (1979/4601) | 1.57 (1.52-1.62) | 1.46 (1.41-1.50) | |
| *Myocardial infarction* [b] | 40.1 (493/1229) | 1.59 (1.49-1.70) | 1.46 (1.36-1.55) | |
| *Cerebrovascular disease* [b] | 58.4 (409/700) | 2.09 (1.96-2.24) | 1.92 (1.80-2.06) | |
| *Peripheral arterial diseases* | 56.5 (386/683) | 1.95 (1.82-2.09) | 1.76 (1.65-1.88) | |
| *Heart failure* [b] | 60.2 (266/442) | 2.10 (1.94-2.27) | 1.83 (1.68-1.98) | |
| *Other CVD* [a] | 37.8 (5053/13372) | 1.34 (1.31-1.37) | 1.29 (1.26-1.32) | |
| No CVD (reference) | 23.5 (33958/144294) | 1 | 1 | |

Model 2 PR (95% CI) (log scale)

Prevalence ratio (PR) and 95% confidence intervals (CIs) for not being in the workforce, [1]Adjusted for age and sex. [2]Adjusted for age, sex, remoteness of residence and education. [a] Based on self-report and hospital records, [b]Based on hospital records only and regardless of presence of other CVD subtypes. Effect sizes were estimated using 'no CVD' as the reference group.

**Fig 2. Non-participation in the workforce: Prevalence and adjusted prevalence ratios in people with and without CVD and according to hospitalisation for CVD subtypes.**

| Subgroup | % (Not in workforce / Total) | | PR (95% CI) | | P-heterogeneity |
|---|---|---|---|---|---|
| | CVD | No CVD | | | |
| **Age group (years)** | | | | | |
| 45-49 | 21.5 (454/2108) | 11.4 (3726/32749) | 1.69 (1.56-1.84) | | <0.0001 |
| 50-54 | 25.7 (968/3764) | 13.7 (5350/39159) | 1.75 (1.65-1.85) | | |
| 55-59 | 35.1 (2001/5706) | 24.0 (9526/39698) | 1.45 (1.40-1.51) | | |
| 60-64 | 56.0 (4234/7559) | 47.0 (15356/32688) | 1.21 (1.19-1.24) | | |
| **Sex** | | | | | |
| Men | 34.8 (3427/9859) | 17.2 (10444/60551) | 1.49 (1.44-1.54) | | <0.0001 |
| Women | 45.6 (4230/9278) | 28.1 (23514/83743) | 1.27 (1.24-1.30) | | |
| **Region** | | | | | |
| Major cities | 36.7 (3478/9465) | 21.8 (16331/74943) | 1.35 (1.31-1.39) | | 0.2172 |
| Inner regional | 44.0 (3034/6902) | 26.1 (13040/50027) | 1.35 (1.31-1.39) | | |
| More remote | 43.1 (1021/2367) | 24.9 (4075/16379) | 1.42 (1.35-1.50) | | |
| **Marital status** | | | | | |
| Not currently married/defacto | 49.8 (2198/4413) | 28.0 (8130/29000) | 1.44 (1.39-1.49) | | <0.0001 |
| Married/defacto | 37.1 (5415/14602) | 22.4 (25639/114484) | 1.32 (1.29-1.35) | | |
| **Highest Education** | | | | | |
| No school certificate | 63.4 (1556/2455) | 46.0 (5146/11196) | 1.34 (1.29-1.38) | | 0.0013 |
| Certificate/diploma/trade | 41.1 (5039/12260) | 25.0 (22488/89841) | 1.39 (1.36-1.42) | | |
| Tertiary | 22.4 (943/4204) | 13.9 (5824/41835) | 1.25 (1.18-1.32) | | |
| **Language other than English** | | | | | |
| Yes | 41.8 (678/1621) | 26.0 (3751/14422) | 1.41 (1.32-1.50) | | 0.2953 |
| No | 39.8 (6979/17516) | 23.3 (30207/129871) | 1.36 (1.33-1.39) | | |
| **County of Birth** | | | | | |
| Australia/NZ | 39.6 (6035/15224) | 23.0 (25858/112574) | 1.36 (1.34-1.39) | | 0.7454 |
| Others | 41.2 (1553/3769) | 25.4 (7838/30898) | 1.35 (1.30-1.41) | | |
| **Alcohol consumption** | | | | | |
| Non-drinkers | 51.0 (3366/6597) | 31.3 (13050/41644) | 1.36 (1.32-1.39) | | 0.4987 |
| Moderate drinkers | 33.6 (3102/9228) | 19.9 (15673/78804) | 1.33 (1.29-1.37) | | |
| Heavy drinkers | 33.6 (1002/2980) | 20.2 (4392/21784) | 1.33 (1.25-1.40) | | |
| **Smoking status** | | | | | |
| Current | 52.6 (1076/2044) | 30.4 (4059/13345) | 1.48 (1.41-1.55) | | <0.0001 |
| Past | 40.8 (3192/7827) | 23.4 (11114/47559) | 1.38 (1.34-1.42) | | |
| Never | 36.5 (3348/9184) | 22.5 (18650/82926) | 1.28 (1.25-1.32) | | |
| **BMI (kg/m2)** | | | | | |
| Underweight (<18) | 56.2 (95/169) | 33.4 (456/1367) | 1.42 (1.22-1.65) | | 0.0389 |
| Normal weight (18–<25) | 37.3 (1733/4650) | 22.4 (11221/50018) | 1.32 (1.27-1.37) | | |
| Overweight (25–<30) | 35.8 (2490/6947) | 21.4 (11155/52081) | 1.32 (1.27-1.36) | | |
| Obese ((30+) | 45.6 (2736/6003) | 27.1 (8449/31132) | 1.40 (1.36-1.45) | | |
| **Medical History: Cancer** | | | | | |
| No | 38.3 (6216/16224) | 22.7 (29227/128889) | 1.35 (1.33-1.38) | | 1.0000 |
| Yes | 49.5 (1441/2913) | 30.7 (4731/15405) | 1.35 (1.30-1.41) | | |
| **Medical History: Diabetes** | | | | | |
| No | 37.4 (6126/16389) | 22.7 (31022/136501) | 1.32 (1.29-1.34) | | 0.5136 |
| Yes | 55.7 (1531/2748) | 37.7 (2936/7793) | 1.34 (1.29-1.40) | | |
| **Medical History: Osteoarthritis** | | | | | |
| No | 38.7 (7053/18220) | 23.0 (32236/140274) | 1.35 (1.32-1.38) | | 0.8192 |
| Yes | 65.9 (604/917) | 42.8 (1722/4020) | 1.34 (1.26-1.42) | | |

0.5 1.0 2.0

**PR (95% CI) on log scale**

Prevalence ratio (PR) and 95% confidence intervals (CIs) for not being in the workforce by those with CVD compared to those without CVD, adjusted for age, sex, remoteness of residence and education.

**Fig 3. Prevalence ratio of not in workforce in population subgroups based on socio-demographic and health related factors.**

16% of those without CVD; among participants with severe functioning limitations, 73% of those with CVD, and 60% of those without CVD, were not working. After adjustment for

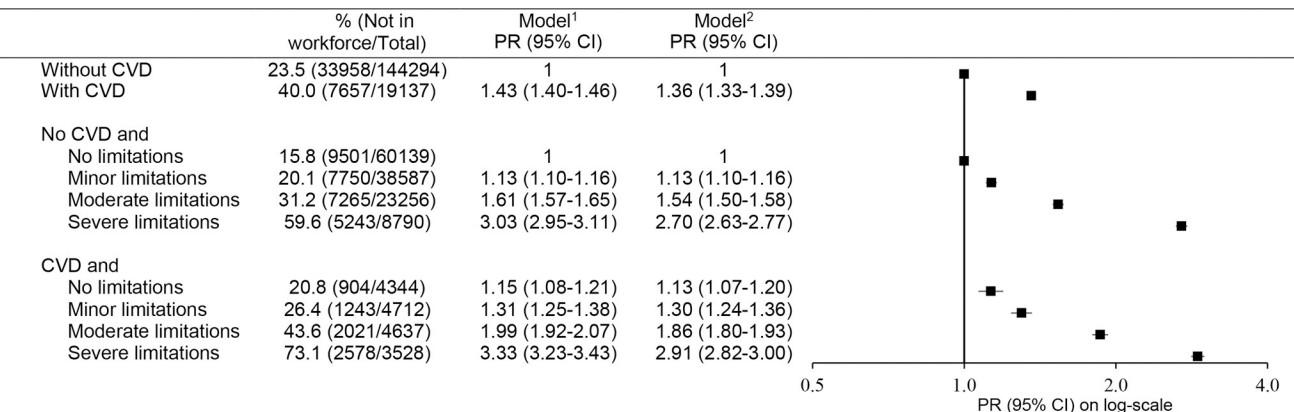

| | % (Not in workforce/Total) | Model[1] PR (95% CI) | Model[2] PR (95% CI) |
|---|---|---|---|
| Without CVD | 23.5 (33958/144294) | 1 | 1 |
| With CVD | 40.0 (7657/19137) | 1.43 (1.40-1.46) | 1.36 (1.33-1.39) |
| | | | |
| No CVD and | | | |
| No limitations | 15.8 (9501/60139) | 1 | 1 |
| Minor limitations | 20.1 (7750/38587) | 1.13 (1.10-1.16) | 1.13 (1.10-1.16) |
| Moderate limitations | 31.2 (7265/23256) | 1.61 (1.57-1.65) | 1.54 (1.50-1.58) |
| Severe limitations | 59.6 (5243/8790) | 3.03 (2.95-3.11) | 2.70 (2.63-2.77) |
| | | | |
| CVD and | | | |
| No limitations | 20.8 (904/4344) | 1.15 (1.08-1.21) | 1.13 (1.07-1.20) |
| Minor limitations | 26.4 (1243/4712) | 1.31 (1.25-1.38) | 1.30 (1.24-1.36) |
| Moderate limitations | 43.6 (2021/4637) | 1.99 (1.92-2.07) | 1.86 (1.80-1.93) |
| Severe limitations | 73.1 (2578/3528) | 3.33 (3.23-3.43) | 2.91 (2.82-3.00) |

Prevalence ratio (PR) and 95% confidence intervals (CIs) for not being in the workforce, [1]Adjusted for age and sex, [2]Further adjusted for remoteness of residence and education attainment. Those with 'no functional limitations and no CVD' were the reference group for estimating prevalence ratios (PR's) for non-participation in work according to joint categories of physical functioning limitations and CVD. CVD is based on both self-report and hospitalisation records. Physical functional limitations had scores ranged from 0 to 100, where higher scores represented fewer limitations, and were grouped into four categories: severe (0–<60); moderate (60-<90), minor (90-<100) and no (100) functional limitation.

**Fig 4. Non-participation in the workforce: Prevalence and adjusted prevalence ratios according to joint categories of physical functioning limitations and CVD.**

sociodemographic variables, compared to those without CVD and no functional limitations, participants without physical functional limitations but with CVD were 13% more likely to be out of the workforce (PR = 1.13, 95%CI = 1.07–1.20). Those with severe functioning limitations were 3 times as likely to be out of the workforce if they had CVD [PR = 2.91 (95% CI: 2.82–3.00)] and 2.7 times as likely if they did not have CVD [PR = 2.70 (95% CI: 2.63–2.77)] (**Fig 4**).

## Discussion

In this large population-based Australian study of older working-age adults, people with CVD were around 36% more likely than those without CVD to be out of the workforce. Non-participation was highest for people with cerebrovascular disease, peripheral arterial disease and heart failure and lowest for other types of CVD. Lower workforce participation among people with CVD was evident regardless of age, sex and other sociodemographic and health-related characteristics. Importantly, physical functioning was strongly related to workforce participation, and in people without physical functioning limitations, participation rates in those with and without CVD were similar, while people with CVD and severe physical functioning limitations were almost three times as likely as those without CVD and no functional limitations to not to be in paid work.

We know of no other studies in Australia that have compared workforce participation in people with and without CVD using large-scale population data. Our study results are broadly comparable to studies set in Canada, Japan, and Europe, which have consistently shown that people with CVD are less likely to participate in the workforce than people without CVD [12, 23, 39]. It is somewhat difficult to compare the magnitude of our findings with these studies, given the variation in study design, definition of workforce participation, case definition of CVD, and selection of the comparison population. For example, a cross-sectional investigation with a study population from 10 European countries found that people with stroke had a 11% higher odds of being unemployed [23] compared to those without stroke. A study with participants in France with stroke were 50% more likely to be out of workforce [12]. The magnitude

of the association between stroke and workforce participation from these European studies was lower than the 92% increase in odds of being out of the workforce for those with vs. without cerebrovascular disease in our study.

We could not find any other study that reported workforce participation in absolute and relative terms among different population sub-groups based on sociodemographic and health-related factors. Similar to a previous report [23], our results indicate that women were more likely to be out the paid workforce compared to men in absolute terms, but the magnitude of the relative association between CVD and workforce participation was greater among men than women. We could not find other study to compare our findings that people with CVD who were single, had education less than tertiary education or were current smokers were more likely in absolute and relative terms to be out of the workforce compared to those without CVD. However, there were some studies to compare the secondary outcomes in our investigations but those varied by countries of the previously reported studies. For example, participants in Italy with MI had 50% higher chance to retire early [24], compared to 27% in our study. Previous studies from 10 European countries have reported on the extent on early retirement of people with CVD subtype such as stroke [23] but comparison could not be made due to variation in the definition of the CVD subtype as indicated in this study.

While our study, like others, highlights the relatively high prevalence of workforce non-participation among people with CVD and a stronger relationship in younger people is also consistent with the idea that CVD is the driving force, it should not be assumed that CVD is the cause without additional evidence. However, cohort studies in Europe and Canada provide evidence consistent with a casual explanation [8–11, 13, 40, 41]. For example, one study from the Netherlands analysed various exit mechanisms from paid employment (such as, via unemployment benefits, via early retirement benefits etc.) and reported that in comparison to those without CVD, people with incident CVD were more likely to leave paid employment regardless of the exit mechanisms [41]. Nevertheless, the relationship between CVD and workforce participation, particularly when measured in terms of government benefits, is highly specific to the social welfare structure of the countries in which study participants live and work. In other studies, risk of leaving employment increased with severe CVD subtypes such as coronary heart disease, stroke and MI [9–11, 13, 40] which are also broadly similar to those reported in our investigation, especially participants living with multiple CVD subtype. Another underlying reason for higher workforce non-participation in people with CVD could be physical disability since physical functioning has been shown to decline following incident CVD [41] and this study has indicated that those with higher physical functioning limitations had the highest probability of being out of paid-work.

This key finding, that impaired physical functioning is likely to be an important factor underpinning the difference in workforce participation rates between those with and without CVD, has not been reported previously. However, it is consistent with the evidence from a European study reporting a relatively high proportion of people with CVD leave the paid workforce via disability pension [42] compared those without CVD. This may also explain the lower participation rates among people with cerebrovascular disease, peripheral artery disease and heart failure compared to those with ischemic heart disease. This is an important finding as cardiac rehabilitation programs help improve physical functional limitations [43] and hence may improve return to work.

Our large-scale population-based study with linkage to hospital records allowed comparison of workforce non-participation in people with and without CVD, within population sub-groups and across CVD subtypes enabling a comprehensive comparative description of workforce participation in individuals living with CVD in the community. However, there was a lack of information on the types of work people were engaged in. This would have been

valuable in better understanding the observed workforce participation patterns. Our study population was randomly sampled from a whole-of-population database, and included ~ 10% of the entire population in the target age-group and the response rate was ~18%, consistent with cohort studies of this nature. Generally, participants in cohort studies are healthier than the general population [44]. Though we could not find a comparable age-group similar ours for prevalence of CVD in Australia, the workforce participation rate in our study was 9% higher (74.6% vs 65.2%), than that reported for Australia for the same age group during the same period (2007–08) by the Australian Bureau of Statistics [45, 46]. Hence, while our absolute estimates of CVD prevalence and workforce participation may not be directly representative, PRs, which are based on internal comparisons, are still likely to be generalisable [47].

More precise point estimates of the workforce participation among people with CVD in Australia may be derived from linked administrative data, which do not rely on survey participation. In future, longitudinal investigation of change in workforce participation in Australia after incident CVD would be valuable for enhancing our understanding about the transitions in workforce participation in relation to major CVD events. Our results show the majority of people with CVD were still in the paid workforce but people who have experienced a CVD event, especially those with physical disability, were more vulnerable to not being in the workforce compared to their counterparts. Given the important of paid work for mental health and overall economy [15], policies aiming to help people with CVD or other major illness remain or re-enter the workforce could help lessen this gap. Thus, our findings will be valuable for healthy and successful aging for those with CVD.

## Conclusion

While the majority of people with CVD continue to undertake paid work, participation is reduced compared to people without CVD. Lower participation was observed for different CVD subtypes and among different population sub-groups, and workforce non-participation was relatively higher for people who had had a stroke, heart failure or peripheral vascular disease, who were in their 50s. Physical functioning limitations was a key factor in non-participation in the paid workforce both for those with and without CVD. Similar associations were observed for weekly paid work hours among those who had been working, retirement among all study participants and retirement due ill health among retirees who had not been working. The results of this investigation have significant policy implications for healthy and successful ageing for those living with CVD. It underpins the importance of rehabilitation and suggests that social policies to encourage employment among older persons should integrate consideration of the role of chronic disease, including CVD.

## Supporting information

**S1 File.**
(DOCX)

## Acknowledgments

This research was completed using data collected through the 45 and Up Study (www.saxinstitute.org.au). The 45 and Up Study is managed by the Sax Institute in collaboration with major partner Cancer Council NSW; and partners: the National Heart Foundation of Australia (NSW Division); NSW Ministry of Health; NSW Government Family & Community Services–Ageing, Carers and the Disability Council NSW; and the Australian Red Cross Blood Service. We thank the many thousands of people participating in the 45 and Up Study.

## Author Contributions

**Conceptualization:** Muhammad Shahdaat Bin Sayeed, Grace Joshy, Emily Banks, Rosemary Korda.

**Formal analysis:** Muhammad Shahdaat Bin Sayeed.

**Investigation:** Muhammad Shahdaat Bin Sayeed.

**Methodology:** Muhammad Shahdaat Bin Sayeed, Grace Joshy, Ellie Paige, Emily Banks, Rosemary Korda.

**Supervision:** Grace Joshy, Ellie Paige, Emily Banks, Rosemary Korda.

**Writing – original draft:** Muhammad Shahdaat Bin Sayeed.

**Writing – review & editing:** Muhammad Shahdaat Bin Sayeed, Grace Joshy, Ellie Paige, Emily Banks, Rosemary Korda.

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
