## [Decision Letter · Decision Letter 0]

25 Nov 2020

PONE-D-20-33041

Cardiovascular disease subtypes, physical disability and workforce participation: a cross-sectional study of 163,562 middle-aged Australians

PLOS ONE

Dear Dr. Bin Sayeed,

Thank you for submitting your manuscript to PLOS ONE. After careful consideration, we feel that it has merit but does not fully meet PLOS ONE’s publication criteria as it currently stands. Therefore, we invite you to submit a revised version of the manuscript that addresses the points raised during the review process. You find our reviewers' comments below.

We look forward to receiving your revised manuscript.

Kind regards,

Thomas Behrens

Academic Editor

PLOS ONE

'None'

**Comments to the Author**

1. Is the manuscript technically sound, and do the data support the conclusions?

Reviewer #1: Yes

Reviewer #2: Yes

2. Has the statistical analysis been performed appropriately and rigorously? 

Reviewer #1: Yes

Reviewer #2: Yes

3. Have the authors made all data underlying the findings in their manuscript fully available?

Reviewer #1: Yes

Reviewer #2: No

4. Is the manuscript presented in an intelligible fashion and written in standard English?

Reviewer #1: Yes

Reviewer #2: No

5. Review Comments to the Author

Reviewer #1: The authors examined the relationship of cardiovascular disease to workforce participation in older working-age people. For this purpose they used questionnaire data from a very large population based study assessed between 2006 and 2009 in New South Wales (Australia) and linked it to hospitalisation data. The study size of more than 160,000 subjects is enormous. The study is comprehensibly described and written in good English. Nevertheless, I do have some comments/questions:

Major

1. How do the authors explain the observation that older people (60-64 years) have lower PRs than people aged 50-55 years (see Fig 3)?

2. The data analysed are relatively old. How do the authors assess the transferability of the results to the present day? Could the results be influenced by the financial crisis of 2007-2008?

3. I stumble upon the number of subjects included. Perhaps the relevant section could be made clearer:

• In the abstract the subjects with CVD (19,137) and without CVD (144,294) do not add up to 163,562.

• Please give numbers of excluded subjects, e.g. for the retirees who had been classified as participating in the workforce (line 117).

• How large was the data set finally evaluated?

4. The references should be revised. For example, in some references the PID is given in others it is missing.

Minor

• Should „Language other than English (LOTE)“ describe the mother tongue? (Line 144) Please specify.

• Why were Australia and New Zealand combined when considering the place of birth (Line 144)?

• Please explain the basis for the categorisation of alcohol consumption. How were the cut-offs chosen? How was an alcoholic drink defined? (Lines 146-148)

• How were the cut-offs chosen for physical functioning limitations? (Lines 153-156)

• The information on the exclusion of persons without data regarding workforce participation should be given in the method section. Please specify how many subjects were excluded. (Lines 173/174)

• Table 1: What does „/“ stand for in the 3rd line (Percentage (%))?

• Inaccuracies in Table 2: It should be „[inter quartile range]“ and „To care for family“ should be shown left centred.

• Fig 1: Why do you mention SEM but do not show SEM?

Reviewer #2: Overall, the manuscript provides valuable information on CVD and workforce participation, based on a large sample size.

Please consider following comments:

1. The response rate should already be mentioned in the methods, briefly, and - if applicable - further aspects with regard to sample composition (weighting?, exclusions due to implausible questionnaires?).

2. Outcome measurement:

Full retirement definition was based on giving reasons for retirement, why not just use the information on current work status (“Completely retired/pensioner”)?

Current work status was indicated by multiple choice, did multiple choices occur and how was this handled then?

Participation in work was also defined by paid work hours including 0 hours?

3. I could not find the number of excluded participants with missing data on workforce participation (either text or flow chart).

4. There seems to be some truncated text in the two tables.

5. The overlap of workforce participation/retirement should not be considered in analysis retirement (line 116). Do the precentages in line 210/11 refer to this, though? Then it could also be included in table 2.

6. Line 281: The "and" in the middle should be "or"?

7. As the authors correctly state in the methods, causal relationships cannot be assessed in this study, though implicitly assumed at least in conclusions. Lines 289 ff seem to discuss possible causal relationships, but the paragraph needs more comprehensible and clear formulation. This should also include physical disablity.

8. I am not familiar with "PRs, which are based on internal comparisons" (line 324), what does this mean/what would be the opposite? Despite the reference44, I doubt that it is correct to negate any selection bias in the study. As education was used in the analysis (and is usually associated with CVD), how comparable was it to the general population?

9. As results for physical disablity are a "key finding", they should also be included in the conclusions.

10. "The results of this investigation have significant policy implications for healthy and successful ageing for those living with CVD." (line 339ff) I cannot follow this: What are the implications for health for people with CVD here?

6. PLOS authors have the option to publish the peer review history of their article (what does this mean?). If published, this will include your full peer review and any attached files.

Reviewer #1: No

Reviewer #2: No

---

## [Author Response · Author response to Decision Letter 0]

9 Mar 2021

RESPONSE TO REVIEWERS’ COMMENTS

EDITOR'S COMMENTS

We have revised the manuscript to confirm with the requirement of the journal.

We have added one paragraph in the cover letter to address the issue of data availability, including the restrictions on sharing a de-identified data, and the ways to access the data.

'None'

In the cover letter above, we have stated about the grants received by authors, and declared that the authors have no other competing interests.

Reviewer #1: 

The authors examined the relationship of cardiovascular disease to workforce participation in older working-age people. For this purpose they used questionnaire data from a very large population based study assessed between 2006 and 2009 in New South Wales (Australia) and linked it to hospitalisation data. The study size of more than 160,000 subjects is enormous. The study is comprehensibly described and written in good English. Nevertheless, I do have some comments/questions:

We thank the reviewer for recognising the importance of this work.

Major

1. How do the authors explain the observation that older people (60-64 years) have lower PRs than people aged 50-55 years (see Fig 3)?

The higher PR in the younger age group is not surprising. The PR is the prevalence (proportion) in the exposed divided by the prevalence in the unexposed (baseline) group. Thus, all else being equal, as the prevalence in the baseline group increases, the PR will decrease. Because the prevalence of (adverse) outcomes commonly increase with age, it is very common to see PRs decrease with age, as is the case here.

Crude prevalence (proportion) of workforce non-participation was higher in 60-64 year olds compared to 50-54 year olds, irrespective of CVD status; see the following extract from Fig 3.

Age-group Proportions (%) not in work PR (95% CI)

 CVD group No CVD group 

50-54 year 25.7 13.7 1.75 (1.65-1.84)

60-64 year 56.0 47.0 1.21 (1.19-1.24)

The following statement has been included in line 253 to 254 in the manuscript:

Although the crude prevalence of workforce non-participation was higher in older compared to younger age groups (irrespective of CVD status), the relation of CVD and workforce non-participation became weaker with increasing age.

2. The data analysed are relatively old. How do the authors assess the transferability of the results to the present day? Could the results be influenced by the financial crisis of 2007-2008?

Population distribution and cardiovascular disease profile in Australia have not drastically changed since these data were collected. Notably, the effect of the financial crisis of 2007-2008 on Australia has been considerably less than in many other countries (ABS, 2010). Regardless, the relative estimates (PRs) comparing those CVD vs without may well be transferable (Rothman et al., 2013). This is particularly so if the financial crisis or other factors have not affected workforce participation disproportionately based on CVD status, but this is unlikely.

Reference:

ABS (Australian Bureau of Statistics), The Global Financial Crisis and its impact on Australia, Year Book Australia 2009–10, Number 91, ABS Canberra, ABS Catalogue No. 1301.0, Chapter 27 — Financial system, 2010, p. 687-88.

Rothman KJ, Gallacher JE, Hatch EE. Why representativeness should be avoided. Int J Epidemiol. 2013 Aug;42(4):1012-4. doi: 10.1093/ije/dys223. PMID: 24062287

3. I stumble upon the number of subjects included. Perhaps the relevant section could be made clearer:

• In the abstract the subjects with CVD (19,137) and without CVD (144,294) do not add up to 163,562.

Thank you for identifying this. The final number of eligible participants was 163,562; this is the number of participants included in the descriptive tables. As outcome-specific exclusions for missing or invalid data were applied for each outcome, the number of participants contributing to regression models were slightly different: 163,431 for workforce participation, 114,064 for paid hours of work and 11,961 for retirement due to ill health. However, there was no missing value for retirement. We have modified the abstract, methods and study flow chart (Fig S1) to make this clearer. 

 

• How large was the data set finally evaluated?

We have modified the abstract, methods and study flow chart (Fig S1) to make this clearer.

4. The references should be revised. For example, in some references the PID is given in others it is missing.

We have checked all references again, aiming to provide PIDs. However, not all references had a PID, hence it was not possible to provide a PID for all references. In the first draft, we used Endnote as reference manager but in this revised version the references were edited manually after unlinking those from the main body of the article.

Minor

• Should ‘Language other than English (LOTE)’ describe the mother tongue? (Line 144) Please specify.

The questionnaire asked ‘Do you speak a language other than English at home’ followed by two options to choose from (yes and no). To further clarify, the wording has been modified both in the text and in the table to ‘language spoken at home other than English (yes/no)’. 

• Why were Australia and New Zealand combined when considering the place of birth (Line 144)?

New Zealand citizens do not need a visa to live or work in Australia and have access to most of the same entitlements, including health and welfare, and opportunities as Australian citizens. Hence, we grouped Australian and New Zealand citizens as one category and classified participants from other countries as others. 

• Please explain the basis for the categorisation of alcohol consumption. How were the cut-offs chosen? How was an alcoholic drink defined? (Lines 146-148)

This variable was categorised with cut points (0, 1-14 and ≥15) aimed to align with the Australian alcohol consumption guidelines to minimise the risk of long-term harm. At the time of data collection, the guideline for low-risk alcohol consumption in Australia was daily consumption of 20 gram of alcohol and the standard drink was defined as 10 gram of alcohol. This meant 14 standard drinks per week was considered low-risk alcohol drink for both men and women in Australia at the time of data collection. The following text with reference is added in the ‘Other variables of interest section’ of main text of the manuscript.

The cut-points of alcohol consumption broadly reflect Australian guidelines on low-risk consumption [33]. 

References

33. NHMRC (National Health and Medical Research Council). Australian Guidelines To Reduce Health Risks From Drinking Alcohol (NHMRC, Canberra, 2009).

• How were the cut-offs chosen for physical functioning limitations? (Lines 153-156)

We have added a reference for the choice of cut points for physical functioning limitations. The manuscript now reads:

Physical functioning limitations scores ranged from 0 to 100, with higher scores representing fewer limitations, and were grouped into four categories: no limitation (score of 100); minor limitation (score 90–99); moderate limitation (60–89); and severe limitation (score 0–59) [35]

35. Stewart A, Kamberg C. Measuring functioning and well-being: the Medical Outcomes Study Approach.Durham, North Carolina: Duke University Press; 1992

Further details are provided in Supplementary Table S6. 

• The information on the exclusion of persons without data regarding workforce participation should be given in the method section. Please specify how many subjects were excluded. (Lines 173/174)

We have updated the methods section of the manuscript to clarify the definition. The following paragraph has been added to the end of ‘Study population and data sources’.

The participants with invalid data on age or date of recruitment (n=454), those with linkage errors (n=195) and those aged 65 years or over (n=102,942) were excluded. There were 163,562 participants who were of working age (45-<65 years old) with known CVD status at baseline (70 458 men, 93 104 female) (details in supplementary file: S1-S4 Figs, S1-S3 Tables).

At the end of ‘Outcome: Workforce participation’ section, the following statement regarding the number of participants excluded is added.

Outcome-specific exclusions for missing or invalid data were applied for each outcome. Of the 163,562 participants included in the study, 131 had missing data on workforce participation status (Supplementary Fig1). Of the 121,816 participants in the work force, 7752 had missing/invalid data on the number of paid hours per week. Of the 41,615 participants not in the work force, 11,961 had invalid data on retirement due to ill health.

• Table 1: What does „/“ stand for in the 3rd line (Percentage (%))?

This section in the table is revised to ‘/163562’ in both columns.

• Inaccuracies in Table 2: It should be „[inter quartile range]“ and „To care for family“ should be shown left centred.

We have updated Table 2.

• Fig 1: Why do you mention SEM but do not show SEM?

The value is very small and thus might not be always visible. A new note ‘‘, and SEM might not be always visible because of small values’’ is embedded in Figure 2. The mean and standard error of mean (SEM) of paid work hours per week according to CVD, age-group are presented for ‘men and women combined’, men and women. These values were used to generate the bar diagram presented in the paper. The mean values were rounded while printing those on top of the figure but while making the bar height, the values upto 5 decimal points were used.

 

Reviewer #2: 

Overall, the manuscript provides valuable information on CVD and workforce participation, based on a large sample size.

We thank the reviewer for recognising the importance of this work.

Please consider following comments:

1. The response rate should already be mentioned in the methods, briefly, and - if applicable - further aspects with regard to sample composition (weighting?, exclusions due to implausible questionnaires?).

A detailed description of the 45 and Up study is available as part of a published protocol (Banks et al,. International journal of epidemiology. 2008;37(5):941-7. doi: 10.1093/ije/dym184) so we did not extensively describe it in this paper. However, we have added in further details on the response rate, weighting and exclusion criteria to the revised manuscript. The newly added texts in the ‘Study population and data sources’ section are as follows:

Individuals joined the study by completing a postal questionnaire (available at https://www.saxinstitute.org.au/our-work/45-up-study/questionnaires/) and providing consent for follow-up through linkage to a range of routinely-collected data. The majority of participants joined in 2008, with the majority completing the baseline questionnaire in February 2008. Persons aged 80 years and over and those living in rural and remote areas were oversampled by a factor of two. The response rate to mailed invitations was estimated to be 18%, with the final sample representing around 10% of the NSW population aged 45 years and older.

The participants with invalid data on age or date of recruitment (n=454), those with linkage errors (n=195) and those aged 65 years or over (n=102,942) were excluded. There were 163,562 participants who were of working age (45-<65 years old) with known CVD status at baseline (70 458 men, 93 104 female) (details in supplementary file: S1-S4 Figs, S1-S3 Tables).

2. Outcome measurement:

Full retirement definition was based on giving reasons for retirement, why not just use the information on current work status (“Completely retired/pensioner”)?

Most participants chose more than one option from the multiple-choice questions that asked about current work status and many were found to be in apparent conflict regarding retirement status. On the other hand, defining retirement status from the questionnaire that asked about the cause of retirement was straight forward. Note we have used the term ‘retired’, and have not differentiated between ‘full-retirement’ or ‘partial retirement’.

Current work status was indicated by multiple choice, did multiple choices occur and how was this handled then? Participation in work was also defined by paid work hours including 0 hours?

Yes, there were participants choosing more than one option from the multiple-choice questions. Logical checks were done using responses to questions on current work status and ‘paid work hours per week’. We have made this clearer in the revised manuscript (first paragraph of the section ‘Outcomes: Workforce participation’ in page 6). The steps of defining workforce participation might explain further how paid work hours per week contributed to the definition of workforce participation, and how inconsistencies between paid work status and paid work hour per week were resolved. The following text is added in the supplementary section in S4 Table.

• Participants were considered to be in paid work if:

o Number of hours of paid work hours is valid (0 to <100) OR 

o Reported being in full time paid work, in part time paid work, self-employed or partially retired.

• Participants were considered to be not in paid work if:

o Number of hours of paid work hours per week is zero (0) or missing (but not invalid) AND

o Not reported being in fulltime paid work, part time paid work, self-employed or partially retired

• People who are not paid for work automatically received zero (0) for paid work hour per week

• For those who are in “paid” category and entered 0 as paid work hour per week, their paid work status was accepted, and weekly paid work hours were invalidated

• For those who are in “not paid” category and entered valid paid work hours per week, their paid work status was changed, and weekly paid work hours were accepted when the weekly paid work hours were larger than 0.

3. I could not find the number of excluded participants with missing data on workforce participation (either text or flow chart).

The proportion is mentioned at the footnote of Table 2. The exact numbers are now mentioned in response to first reviewer’s query number 3 as above and also, in the supplementary file (S1-S4 Figs and S1-S3 Tables).

4. There seems to be some truncated text in the two tables.

We have updated the both tables.

5. The overlap of workforce participation/retirement should not be considered in analysis retirement (line 116). Do the precentages in line 210/11 refer to this, though? Then it could also be included in table 2.

This has been updated and the exact numbers are now mentioned in the text with relevant wordings. It was explained in query 3 of first reviewer, and further details are provided in the supplementary file (S1-S4 Figs and S1-S3 Tables).

6. Line 281: The "and" in the middle should be "or"?

We have updated this sentence and now in line 295 in the clean version.

7. As the authors correctly state in the methods, causal relationships cannot be assessed in this study, though implicitly assumed at least in conclusions. Lines 289 ff seem to discuss possible causal relationships, but the paragraph needs more comprehensible and clear formulation. This should also include physical disablity.

This paragraph has been updated and includes the issue of physical disability as a possible underlying mechanism which is further explained in the paragraph that follows it. The wording is as follows:

Another underlying reason for higher workforce non-participation in people with CVD could be physical disability since physical functioning has been shown to decline following incident CVD [41] and this study has indicated that those with higher physical functioning limitations had the highest probability of being out of paid-work. 

Reference:

41. Kucharska-Newton A, Griswold M, Yao ZH, Foraker R, Rose K, Rosamond W, et al. Cardiovascular Disease and Patterns of Change in Functional Status Over 15 Years: Findings From the Atherosclerosis Risk in Communities (ARIC) Study. J Am Heart Assoc. 2017;6(3):e004144. pmid: 28249844.

8. I am not familiar with "PRs, which are based on internal comparisons" (line 324), what does this mean/what would be the opposite? Despite the reference44, I doubt that it is correct to negate any selection bias in the study. As education was used in the analysis (and is usually associated with CVD), how comparable was it to the general population?

The ratio of prevalences of two groups is called prevalence ratio (PR), and it is analogous to the risk ratio (RR) in cohort studies. Although the absolute proportions estimated in this study will likely not reflect the absolute numbers in the target population, it is reasonable to assume that the relative comparison between groups (in this case the PRs of CVD compared to non-CVD) will be similar in the target population if confounding is minimised (see also response to Reviewer 1, point 2). We have acknowledged the possibility of selection bias but also emphasised the strength of findings from internal comparison in this investigation. The wording of the sentence is now updated as follows:

Hence, while our absolute estimates of CVD prevalence and workforce participation may not be generalisable, PRs, which are based on internal comparisons, are still likely to be generalisable [47].

47. Mealing NM, Banks E, Jorm LR, Steel DG, Clements MS, Rogers KD. Investigation of relative risk estimates from studies of the same population with contrasting response rates and designs. BMC Med Res Methodol. 2010;10:26. pmid: 20356408

We sought to look for other comparable studies but could not find any that investigated the association of CVD, workforce participation and education among study participants similar to ours. 

9. As results for physical disablity are a "key finding", they should also be included in the conclusions.

The issue of physical disability was previously mentioned at the end of the sentence (in line 337 of the old version and line 409 of the new version). Now in this version we have deleted the end section of that sentence and added a new sentence stating it as a key finding in the conclusion (in line 362-363 of new clean version), which is as follows. 

Physical functioning limitations was a key factor in non-participation in the paid workforce both for those with and without CVD.

10. "The results of this investigation have significant policy implications for healthy and successful ageing for those living with CVD." (line 339ff) I cannot follow this: What are the implications for health for people with CVD here?

The following text is added at the end of the discussion section to clarify the implications (last section of the final paragraph of the discussion section). 

Our results show the majority of people with CVD were still in the paid workforce but people who experienced a CVD event, especially those with physical disability, were more vulnerable to not being in the workforce compared to their counterparts. Given the importance of paid work for individual wellbeing and the overall economy [15], policies aimed at helping people with CVD, particularly those with accompanying physical disability, to remain or re-enter the workforce would be of benefit to both individuals and society. 

Reference:

15. Schofield D, Shrestha R, Percival R, Passey M, Callander E, Kelly S. The personal and national costs of CVD: impacts on income, taxes, government support payments and GDP due to lost labour force participation. Int J Cardiol. 2013;166(1):68-71. pmid: 22018513. 

EDITOR'S COMMENTS

PLOS authors have the option to publish the peer review history of their article (what does this mean?). If published, this will include your full peer review and any attached files.

We agree to publish the full peer review history.

With best wishes,

Yours sincerely,

Muhammad Shahdaat Bin Sayeed, on behalf of the authors

ORCID ID: 0000-0003-0027-9614

---

## [Decision Letter · Decision Letter 1]

24 Mar 2021

Cardiovascular disease subtypes, physical disability and workforce participation: a cross-sectional study of 163,562 middle-aged Australians

PONE-D-20-33041R1

Dear Dr. Bin Sayeed,

We are pleased to inform you that your manuscript has been judged scientifically suitable for publication and will be formally accepted for publication once it meets all outstanding technical requirements.

Kind regards,

Thomas Behrens

Academic Editor

PLOS ONE

Additional Editor Comments (optional):

Reviewers' comments:

Reviewer's Responses to Questions

**Comments to the Author**

1. If the authors have adequately addressed your comments raised in a previous round of review and you feel that this manuscript is now acceptable for publication, you may indicate that here to bypass the “Comments to the Author” section, enter your conflict of interest statement in the “Confidential to Editor” section, and submit your "Accept" recommendation.

Reviewer #1: All comments have been addressed

Reviewer #2: All comments have been addressed

2. Is the manuscript technically sound, and do the data support the conclusions?

Reviewer #1: Yes

Reviewer #2: Yes

3. Has the statistical analysis been performed appropriately and rigorously? 

Reviewer #1: Yes

Reviewer #2: Yes

4. Have the authors made all data underlying the findings in their manuscript fully available?

Reviewer #1: Yes

Reviewer #2: Yes

5. Is the manuscript presented in an intelligible fashion and written in standard English?

Reviewer #1: Yes

Reviewer #2: Yes

6. Review Comments to the Author

Reviewer #1: I enjoyed reading the revised manuscript and the answers to my questions. All my questions were answered to my complete satisfaction. I have no further comments.

I thank the authors for this valuable work.

Reviewer #2: All comments have been adressed. An additional minor recommendation is to use thousands seperators in the tables, too.

7. PLOS authors have the option to publish the peer review history of their article (what does this mean?). If published, this will include your full peer review and any attached files.

Reviewer #1: No

Reviewer #2: No

---

## [Editor Report · Acceptance letter]

30 Mar 2021

PONE-D-20-33041R1 

Cardiovascular disease subtypes, physical disability and workforce participation: a cross-sectional study of 163,562 middle-aged Australians 

Dear Dr. Bin Sayeed:

I'm pleased to inform you that your manuscript has been deemed suitable for publication in PLOS ONE. Congratulations! Your manuscript is now with our production department. 

Kind regards, 

on behalf of

Prof. Thomas Behrens 

Academic Editor

PLOS ONE